# In Vitro Assessment of Hydrolysed Collagen Fermentation Using Domestic Cat (*Felis catus*) Faecal Inocula

**DOI:** 10.3390/ani12040498

**Published:** 2022-02-17

**Authors:** Christina F. Butowski, David G. Thomas, Nick J. Cave, Emma N. Bermingham, Douglas I. Rosendale, Shen-Yan Hea, Halina M. Stoklosinski, Wayne Young

**Affiliations:** 1Smart Foods, AgResearch Limited, Te Ohu Rangahau Kai, Palmerston North 4474, New Zealand; nina.butowski@agresearch.co.nz (C.F.B.); wayne.young@agresearch.co.nz (W.Y.); 2Centre for Feline Nutrition, Massey University, Palmerston North 4442, New Zealand; d.g.thomas@massey.ac.nz; 3School of Agriculture and Environment, Massey University, Palmerston North 4474, New Zealand; 4School of Veterinary Science, Massey University, Palmerston North 4474, New Zealand; n.j.cave@massey.ac.nz; 5Anagenix, Auckland 1052, New Zealand; doug.rosendale@anagenix.com; 6Digital Agriculture, AgResearch Limited, Grasslands, Palmerston North 4410, New Zealand; shen.hea@agresearch.co.nz; 7The New Zealand Institute for Plant and Food Research Limited, Fitzherbert Science Centre, Palmerston North 4474, New Zealand; halina.stoklosinski@plantandfood.co.nz

**Keywords:** in vitro, fermentation, butyrate, feline, collagen, faecal donor

## Abstract

**Simple Summary:**

The gastrointestinal microbiome of domestic cats can utilise dietary fibre to produce beneficial fermentation end products such as butyrate, an energy source for intestinal cells. However, domestic cats are obligate carnivores and therefore may not require plant derived dietary fibre. It has been hypothesised that in the wild, consumption of animal-derived substrates such as hair and cartilage may be fermented by the cat’s gastrointestinal microbiome, producing butyrate and other beneficial end products. Therefore, the aim of this study was to use a laboratory digestion and fermentation model (simulating the digestion process) to determine the concentrations of butyrate produced by various animal-derived substrates, including hydrolysed collagen, hair, and cartilage. Faecal samples were used in batch cultures to represent the microbes in the cat’s colon. Faecal samples came from cats fed either high protein or high carbohydrate diets. The type of faeces used was found to affect the fermentation profile produced, likely due to differing microbial community compositions, at the start of the batch cultures. Microbes present in high protein faeces fermented the hydrolysed collagen substrate, producing increased concentrations of butyrate. This finding has wider implications for use of animal-derived substrates in species-appropriate diet formulations.

**Abstract:**

The gastrointestinal microbiome has a range of roles in the host, including the production of beneficial fermentation end products such as butyrate, which are typically associated with fermentation of plant fibres. However, domestic cats are obligate carnivores and do not require carbohydrates. It has been hypothesised that in the wild, collagenous parts of prey—the so-called animal-derived fermentable substrates (ADFS) such as tendons and cartilage—may be fermented by the cat’s gastrointestinal microbiome. However, little research has been conducted on ADFS in the domestic cat. Faecal inoculum was obtained from domestic cats either consuming a high carbohydrate (protein:fat:carbohydrate ratio of 35:20:28 (% dry matter basis)) or high protein (protein:fat:carbohydrate ratio of 75:19:1 (% dry matter basis)) diet. ADFS (hydrolysed collagen, cat hair, and cartilage) were used in a series of static in vitro digestions and fermentations. Concentrations of organic acids and ammonia were measured after 24 h of fermentation, and the culture community of microbes was characterised. The type of inoculum used affected the fermentation profile produced by the ADFS. Butyrate concentrations were highest when hydrolysed collagen was fermented with high protein inoculum (*p* < 0.05). In contrast, butyrate was not detectable when hydrolysed collagen was fermented in high carbohydrate inoculum (*p* < 0.05). The microbiome of the domestic cat may be able to ferment ADFS to provide beneficial concentrations of butyrate.

## 1. Introduction

Domestic cats are obligate carnivores and do not require carbohydrates to fulfil their energy requirements [1]. However, consumption of dietary fibre may be important for their gastrointestinal microbiota, which has been associated with host health, nutrition, and overall wellbeing [2]. It has been hypothesised that in the wild, consumption of the collagenous parts of prey (such as skin, tendons, and cartilage) may act as a form of dietary fibre, termed herein as animal-derived fermentable substrates (ADFS).

The gastrointestinal microbiome of domestic cats is able to utilise dietary fibre (such as inulin) to produce beneficial fermentation end products, namely organic acids such as acetate, butyrate, and propionate (short chain fatty acids, SCFA) [3]. In human and rodent models, butyrate is an energy source for colonocytes and, therefore, greatly beneficial for the host [4,5]. Conversely, ammonia (from bacterial proteolysis) is typically thought to be detrimental to colonocytes if concentrations become too high in the intestinal lumen, as it inhibits mitochondrial oxygen consumption and SCFA oxidation [6,7]. Therefore, it is of interest to understand if the gastrointestinal microbiota can ferment ADFS in a similar manner to dietary fibre and produce similarly beneficial end products.

In vitro digestion and fermentation models are often used to replace or minimise the use of animal models. They have been used to assess fermentation characteristics of dietary fibre [8,9,10] by assessing factors such as gas production and fermentation end products. Research has shown that the source of the faecal inoculum used in in vitro experiments determines the fermentation end products produced because the microbiota present are pre-determined [11]. This is dictated predominantly by the diet consumed by the donor, which determines the microbiota that are able to utilise the substrates provided.

Depauw, et al. [12] assessed the in vitro fermentation of various ADFS (including hydrolysed collagen, rabbit skin, hair, and bone) using faecal inoculum from a cheetah (*Acinonyx jubatus*) that had consumed a whole rabbit diet. They observed that the microbiota present in the faecal inoculum were able to ferment collagen, producing similar amounts of organic acids per gram of organic matter (OM) to those resulting from the fermentation of fructo-oligosaccharide (FOS). This finding introduces the possibility that ADFS may fulfil the role of dietary fibre in obligate carnivores. A recent study by Deb-Choudhury, et al. [13] also showed that a wool hydrolysate could be fermented by the cat (*Felis catus*) in vivo. This suggests that a range of compounds could act as ADFS for the cat.

The aim of this study was to evaluate the fermentability of ADFS, namely, hydrolysed collagen, cartilage (fresh and freeze-dried), and cat hair (intact and chopped). This was carried out through assessment of organic acids produced by fermentation of the ADFS with faecal inoculum for a period of 24 h, which is the average transit time of digesta for young cats [14]. The ADFS were compared against inulin and cellulose as positive and negative controls, respectively. Two sources of faecal inoculum were used: one from donor cats consuming a high protein diet, and the other from donor cats consuming a high carbohydrate diet.

It was hypothesised that ADFS would be fermented by the microbiota present in the faecal inoculum. Furthermore, we hypothesised that microbiota present in the faeces of the cats fed a protein rich diet would be able to metabolise the protein-rich substrates readily, as opposed to the microbiota present in the faeces of cats fed a carbohydrate-rich diet.

## 2. Materials and Methods

This protocol was approved by the Massey University Animal Ethics Committee (MUAEC Protocol 18/08). Two cohorts of cats were used to provide the faecal inoculum for this study. All cats were housed at the Massey Centre for Feline Nutrition (Palmerston North, New Zealand) for the duration of the collection period.

### 2.1. Faecal Inoculum Collection

The cats providing the faecal inoculum were maintained on their nutritionally complete and balanced diets for a minimum of 21 days prior to faecal collection. They were healthy adult domestic short-haired cats of both male and female sexes, and aged between 2–8 years. The cohort (*n* = 8) providing the high-carbohydrate (CD) faecal inoculum was fed a diet with a protein:fat:carbohydrate ratio of 35:20:28 (% DM basis, Nutro™, MARS Petcare, Raglan, NSW, Australia). Cats providing the high protein inoculum (PD) were fed a diet with a protein:fat:carbohydrate ratio of 75:19:10 (% DM basis, Chef, Kraft Heinz Wattie’s, Hastings, New Zealand). Cats were housed in individual cages (80 cm × 80 cm × 110 cm) until defecation occurred. The faeces were collected within ten minutes of voiding, snap-frozen in liquid nitrogen, then stored at −80 °C before use in faecal fermentation studies.

### 2.2. Substrates

A range of substrates was evaluated, including two forms of hydrolysed collagen: Peptan B (‘PHC’; Peptan B 2000 LD: Rosselott, New Zealand) and a hydrolysed bovine skin product (‘AHC’; ANZCO Foods, Christchurch, New Zealand). Individual rings of bovine tracheal cartilage were isolated from the trachea, connective tissue removed, and the cartilage sliced into ~40 mm × 10 mm fragments (‘fresh cartilage’). A subset of the cartilage was minced (Wolfking, Leingarten, Germany) before being freeze-dried (‘freeze-dried cartilage’). Hair obtained from domestic cats as part of normal grooming practices was collected from the Centre for Feline Nutrition (Massey University, Palmerston North). The hair was either left intact (‘intact cat hair’) or chopped up into ~0.5 cm length (‘chopped cat hair’). ‘inulin’ (Orafti Synergy 1^®^, Benuo, Belgium) and ‘cellulose’ (Avicel^®^, Hawkins Watts, New Zealand) were chosen as positive and negative controls, respectively, for fermentation.

### 2.3. In Vitro Digestion

In vitro digestion was undertaken using the static model method published by Minekus, et al. [15] which was modified to represent the physiology of the domestic cat; temperature was adjusted to 39 °C (domestic cat body temperature), the pH of simulated gastric fluid (SGF) was reduced to 2.5 (domestic cat stomach pH), and the oral phase was removed from the study, as cats lack salivary amylase [16].

A single batch was performed, including all substrates. A 5 g aliquot of substrate was added to 10 mL of SGF (Table A1) and 1 mL of 2% pepsin stock (20 mg/mL SGF) in a Schott bottle. Next, 300 mM CaCl_2_ was added, pH was corrected to 2.5 as required and the mixture vortexed for 60 s at 600 rpm to ensure thorough mixing. The bottles were then placed in a shaking incubator (65 rpm) and incubated at 39 °C for two hours. Simulated intestinal fluid (SIF: Table A1) (11 mL at pH 7) was then added along with 2.5 mL of 16 mM bile stock (Sigma-Aldrich, New Zealand), 40 µL of 300 mM CaCl_2_, 3 mL of reverse osmosis (RO) water and approximately 20–40 µL of 1 M sodium hydroxide (NaOH) until the pH was stabilised at 6.5. The mixture was then returned to the shaking incubator for a further 10 min. The bottles were subsequently removed and 5 mL of porcine pancreatin solution (800 U/mL in SIF solution) (Table A1) was added before they were returned to the shaking incubator for a further two hours. To deactivate the enzymes, following incubation, all Schott bottles were microwaved for 90 s, then put on ice.

Retentate was then dialysed as per manufacturer’s instructions (Spectra/Por Dialysis membrane, 100–500 D, Biotech CE Tubing: Pacific Laboratory Products Pty Ltd., Auckland, New Zealand). Briefly, dialysis tubes were then placed in RO water and stored in a cold room at 4 °C, with water changed three times in 24 h. The resulting retentate was removed from the dialysis tubing and stored at −20 °C before use in the in vitro fermentation.

### 2.4. In Vitro Fermentation

Each substrate was fermented in triplicate for 0, 4, 8, 12, and 24 h per faecal inoculum in autoclaved 5 mL Hungate tubes. Tubes were pre-warmed in a shaking incubator at 39 °C.

Defrosted retentate tubes were mixed thoroughly. An aliquot of phosphate buffer (pH 6.8) was added, mixed thoroughly, then bubbled with nitrogen for one minute to remove the dissolved oxygen present. The bottles were left for a further minute before 3% L-cysteine was added.

A 10% faecal solution was prepared from the PD and CD inocula using sodium phosphate buffer and were manually strained through a mesh bag (MicroAnalytix, Auckland, New Zealand). An aliquot of the inoculum to be tested was immediately pipetted into Eppendorf tubes kept on ice to represent the 0-h sample. For the other timepoints, PD or CD inoculum was added to Hungate tubes containing the allocated substrates. Each tube was subsequently topped with a layer of carbon dioxide, capped, and placed in a shaking incubator at 39 °C for either 4, 8, 12, or 24 h. Upon reaching the allocated time point, tubes were removed from the incubator, placed onto ice, vortexed, and centrifuged at 14,000× *g* for 15 min. The resulting supernatant and pellet were frozen immediately at −80 °C.

### 2.5. Laboratory Analysis

#### 2.5.1. Macronutrient Profiles of Substrate

Moisture content of the substrate was determined using a convection oven at 105 °C, and ash residue at 550 °C (AOAC 930.15/925.10/942.10). Crude fat was analysed using the Soxtec 8000 meat extraction methodology (AOAC 991.36). Crude fibre was analysed using the non-enzymatic gravimetric method (AOAC 962.09/978.10). Total, soluble, and insoluble dietary fibre were calculated using the Megaenzyme assay (AOAC 991.43). Gross energy was measured using bomb calorimetry.

Nitrogen was measured using the Dumas method (AOAC 968.06). Typically, a conversion factor of 6.25 is applied to the nitrogen content of a sample to obtain the crude protein content. For the purposes of this study, the conversion factor of gelatin, 5.55, was applied to AHC, PHC (collectively denoted as hydrolysed collagen due to their similar compositions), and cat hair to account for differences in protein content [17] (Table 1). NFE were calculated by difference; 100 − (crude protein + crude fat + crude fibre + ash).

#### 2.5.2. Organic Acids and Ammonia

Supernatant was diluted 1:5 with PBS containing 2-ethylbutyric acid as an internal standard as previously described [18]. Briefly, aqueous extracts were acidified, phase separated into diethyl ether and stored at −80 °C until analysis. Organic acids were derivatised with N-tert-butyldimethylsilyl-N-methyl trifluoroacetamide plus 1% tert-butyldimethylchlorosilane (MTBSTFA + TBDMSCI, 99:1; Sigma-Aldrich, Auckland, New Zealand) and analysed on a Shimadzu capillary GC system (GC-2010 Plus, Tokyo, Japan) equipped with a flame ionization detector (FID) and fitted with a Restek column (SH-Rtx-1, 30 m × 0.25 mm ID × 0.25 µm) using helium as the carrier gas. The GC-FID was controlled, and data processed, using a Shimadzu GC Work Station LabSolutions Version 5.3, with sample organic acids quantified in reference to authentic standards. Ammonia was measured using the phenol-nitroprusside method [19].

#### 2.5.3. 16S rRNA Amplicon Sequencing

Metagenomic DNA was extracted from the pellet using NucleoSpin Soil kits (Macherey-Nagel, Düren, Germany) according to the manufacturer’s instructions, with modifications as follows. The DNA pellet was defrosted and centrifuged at 15,000× *g* for 3 min. Any remaining supernatant was removed and discarded. Lysis buffer and enhancer were added to the DNA pellet and gently mixed, then the mixture was transferred into a bead beating tube and mixed thoroughly using a Mini-Beadbeater-96 (BioSpec Products, Bartlesville, OK, USA) for four minutes.

Microbial profiles were determined by analysis of the V3 to V4 region of the bacterial 16S rRNA gene using Illumina MiSeq paired-end 2 × 250 base pair amplicon sequencing [20]. QIIME 1.8 was used to process sequences, with quality filtering of reads undertaken using the default settings. Forward and reverse reads were subsequently joined using the ‘join_paired_ends’ function [21]. The USEARCH method was used to align reads to the Greengenes database and identify chimeric sequences, which were subsequently removed from further analysis. UCLUST was used to cluster sequences at 97% similarity into OTUs, which were then assigned taxonomy using RDP classifier.

### 2.6. Statistical Analysis

All analyses were completed using R version 3.6.0 [22].

#### 2.6.1. Organic Acids and Ammonia

To determine changes to organic acid and ammonia concentrations over time, triplicate samples were plotted according to substrate by faecal inoculum using the R package ‘ggplot2’. Samples from the 0-h time point were removed from this analysis as there were only six samples at this time point (only from the control group) and, therefore, could not be plotted for each substrate. Scatterplots and mean lines were used to visualise the data. Heptanoate, hexanoate isobutyrate, isovalerate, and valerate were below the limit of detection in all samples; therefore, they were excluded from any analyses.

Samples were split by faecal inoculum (high protein and high carbohydrate), and only the 24-h time point was assessed as the final endpoint of fermentation. Model-based predicted means, standard errors, and post hoc multiple comparisons were calculated using the ‘predictmeans’ package [23]. Multiple comparison *p*-values were adjusted using the “Tukey” method and are presented as letter-based comparisons. Assumptions of homogeneity and normality were met. Unless otherwise stated, data are presented as mean +/− standard error of the mean (SEM).

#### 2.6.2. Bacterial Profile of In Vitro Fermentation

The 221 samples analysed had a mean sequencing depth of 21,206 reads. One sample was removed from the dataset (a 24-h intact cat hair CD) because of extremely low read numbers (<2500 reads), leaving *n* = 111 PD inoculum samples and *n* = 110 CD inoculum samples for analysis. To consolidate the data into phyla and genera, the R mixOmics package was used. Taxa of low relative abundance were removed (taxa present at <0.0005% relative abundance in six or more samples). At the phylum level, a total of 25 phyla were observed before filtering, and seven remained afterwards. Before filtering, a total of 482 bacterial taxa were observed at the genera level in these samples. After filtering rare taxa, 98 genera remained, which provided the dataset for all further analyses. Chao1 diversity index was used to assess alpha diversity with Kruskal–Wallis analysis to assess significant differences between the substrates.

Permutation ANOVA was used to determine changes by ‘Substrate’ and ‘Faecal Inoculum’ as covariates. Interaction between Substrate and Faecal Inoculum were then investigated. FDR < 0.05 was considered statistically significant. The top 30 most relatively abundant taxa present in each substrate at 24 h of in vitro fermentation were then visualised using bar plots in R using the ggplot2 package. Principal component analysis was also used to assess the beta diversity of bacterial genera.

## 3. Results

### 3.1. High Protein Faecal Inoculum (PD)

#### 3.1.1. Organic Acid and Ammonia Profiles

Organic acid profiles of each substrate across the 24-h sampling period of in vitro fermentation are shown in Figure 1A–C. The substrate being fermented affected the concentration of fermentation end products (Table 2). Fermentation of AHC and PHC for 24 h produced the greatest concentration of butyrate in comparison to the other substrates (*p* < 0.05) (Figure 1B). Fermentation of cat hair also produced significantly higher concentrations of butyrate and propionate (*p* < 0.05). In vitro fermentation in the control group produced the highest mean concentration of ammonia (*p* < 0.05); 19.6 ± 0.544 mM (Figure 2). The lowest mean concentration of ammonia was produced by the fermentation of inulin; 8.2 ± 0.178 mM (Table 2).

#### 3.1.2. Culture Community of High Protein Faecal Inoculum

The effect of the substrate on the culture microbial community varied according to the source of the faecal inoculum (Figure A1). Two-way permutation ANOVA of 76 taxa with substrate and inoculum type as factors showed that both these factors had large effects on the microbial community with significant interactions between substrate and inoculum source (FDR < 0.05). Of these 76, taxa with relative abundance <1% across all substrates were filtered out for the purposes of displaying using bar plots, leaving 30 dominant taxa across all samples (Figure 3). *Fusobacterium* was the most dominant genera in AHC (23%) and PHC (33%) samples (Figure 3). *Escherichia–Shigella* was the most relatively abundant taxa (>20%) present in all other substrates (Figure 3). Chao1 alpha diversity assessment observed significant differences between the substrates (*p* = 0.012; Figure A2).

### 3.2. High Carbohydrate Faecal Inoculum (CD)

#### 3.2.1. Organic Acid and Ammonia Profiles

Organic acid profiles of each substrate across the 24-h sampling period of in vitro fermentation are shown in Figure 1A–C. The substrate being fermented affected the concentration of fermentation end products (Table 3). After 24 h, fermentation of chopped and intact cat hair produced the greatest (*p* < 0.05) concentration of butyrate and propionate when compared to the other substrates (Figure 1B,C). Fermentation of inulin produced the highest (*p* < 0.05) concentration of lactate, compared to the other substrates (Appendix A; Table 3), although freeze-dried and fresh cartilage produced significantly greater amounts of lactate than the control group (*p* < 0.05).

In vitro fermentation in the control group and of cat hair substrates produced the highest mean concentration of ammonia (c.17 ± 0.081 mM; *p* < 0.05) (Figure 2). The lowest concentration of ammonia was produced by the fermentation of inulin; 10.8 ± 0.283 mM (*p* < 0.05) (Table 3; Figure 2).

#### 3.2.2. Culture Community of High Carbohydrate Faecal Inoculum

At 24 h, *Escherichia–Shigella* was the taxon with the greatest relative abundance ranging from 17% of sequence reads in the intact cat hair samples, to 55% in the AHC and PHC samples (Figure 4). The next most relatively abundant taxon in the AHC and PHC samples was *Bacteroides* (12% and 7%, respectively). *Bacteroides* also had the second-greatest relative abundance in the chopped and intact cat hair samples (11% and 14%, respectively) followed by *Prevotella 9* (c.7%) (Figure 4). Chao1 alpha diversity assessment observed significant differences between the substrates (*p* < 0.05; Figure A3).

## 4. Discussion

This study aimed to assess a variety of ADFS to determine their fermentation end products, and subsequently identify which of them may confer a beneficial organic acid profile (i.e., increased butyrate and decreased ammonia). All substrates studied, except cellulose, were readily fermented by the feline inoculum. However, as expected, the type of faecal inoculum (PD or CD) had a major impact on the culture communities and fermentation profiles observed. The fermentation of hydrolysed collagen (both AHC and PHC) produced the highest butyrate concentrations, but only when they were fermented with the PD inoculum. In contrast, in the CD inoculum, fermentation of chopped and intact cat hair substrates produced the greatest concentrations of butyrate.

### 4.1. High Protein Faecal Inoculum (PD)

Fermentation of both AHC and PHC substrates significantly increased concentrations of acetate, butyrate, and propionate. Butyrate production was greatest from the fermentation of hydrolysed collagen. This may be because these substrates have a high crude protein content (97% DM in AHC and PHC vs. 90% DM for the other substrates) that can be utilised by *Fusobacterium* [24] and *Escherichia–Shigella* [25] present in high relative abundances in this model. Previous studies have shown that PHC (when fermented in cheetah inoculum) also results in relatively high butyrate concentrations [12].

Both substrate and faecal donor are factors that have previously been shown to affect ammonia production [26]. An in vitro study by Pinna, et al. [27], showed that protein content in the fermentation system was positively correlated with ammonia production. Production of ammonia by *Escherichia–Shigella* [26], *Bacteroides*, and *Clostridium* [28], all of which were enriched in these samples, can occur via the utilisation of both peptides and amino acids; however, peptides have been found to yield greater ammonia concentrations [26,29]. Surprisingly, despite the higher protein content of the AHC and PHC substrates, ammonia production was similar between all substrates that were assessed.

Other markers of protein fermentation (e.g., branched chain fatty acids) were below the limit of detection in this study. This may be due to the low level of valine, leucine, and isoleucine in the ADFS studied [30]. However, other methodological factors such as sampling times or cross-feeding within the system may also explain these results.

### 4.2. High Carbohydrate FAECAL Inoculum (CD)

In this in vitro system, chopped and intact hair were the substrates that produced the highest butyrate concentrations, but this was not observed by Depauw et al. (2012) [12], when fermenting rabbit hair, despite both hair types being structurally similar [31]. Cat hair is primarily composed of keratin, which is not typically readily fermentable, but the addition of the in vitro digestion step in the current study may have altered the keratin structure, allowing its utilisation by bacteria present in the high-carbohydrate faecal inoculum. Therefore, the increased production of butyrate may be due to this step in the experiment, or bacterial cross-feeding in the in vitro system.

### 4.3. Model Considerations

This study provides further insight into the benefits and limitations of a static in vitro model for companion animal research. In vitro models can provide important information and act as a screening process to test and build hypotheses, but it must be emphasised that they do not truly represent the complexity of the cat’s colonic environment. A limitation of in vitro models is the inability to account for absorption of metabolites that would occur in the colon, and the impact of the starter faecal inoculum on in vitro fermentation patterns [11]. While the latter was assessed in the current study, other factors may also influence the results observed, including the time of sampling or the number of biological replicates of donor faeces.

In this study, the 24-h time point was chosen for substrate comparison, given that the total transit time of young cats is approximately 26 h [14]. Previous in vitro studies investigated longer incubation periods (up to 72 h) but observed no further increase in gas production after 24 h, indicating that fermentation had reached capacity and all substrates were fully fermented [12]. In the current study, the substrates may still have been fermenting, but the corresponding microbial profiles indicated an abundance of *Escherichia–Shigella*. Therefore, continuing the fermentation for a longer period (>24 h) may not have provided any further clarification as to which substrate would confer the most beneficial effects in vivo.

As the triplicate measurements at each sampling time point were taken from the same faecal inoculum, they are pseudoreplicates from the same inoculum rather than true biological replicates. However, this is a common limitation that in principle applies to most culture experiments [32]. To better represent the true population, more faeces from more cats would be required. However, previous in vitro studies also used a pooling technique from a limited number of animals (2–4 faecal samples) [9,27,33,34]. More recently, Bosch, et al., [35] studied the impacts of the number of faecal samples and donors, and observed that the degree of inter-individual variation was dependent on the complexity of the substrate.

Nevertheless, a major advantage of an in vitro model is the ability to screen and document changes over time for multiple substrates simultaneously. By determining changes to organic acid and ammonia concentrations in this time period, comparative information can be generated which would not be possible in vivo. This model successfully screened substrates according to their fermentative properties, identifying a substrate which may confer beneficial effects when used in conjunction with a complete and balanced raw meat diet in vivo. Additionally, the findings in this study highlight the potential use of ADFS in species-appropriate diet formulations for domestic cats that do not have a carbohydrate requirement.

## 5. Conclusions

This study used an in vitro model to determine changes to the fermentation end products and bacterial profiles from a variety of ADFS. ADFS were indeed fermented, but the starting faecal inoculum had the greatest effect on the fermentation profiles observed. The faecal inoculum from cats fed with high protein diets was able to readily ferment high protein substrates due to the bacteria present in the culture community. This finding has wider implications for the use of ADFS in species-appropriate diet formulations.

## Figures and Tables

**Figure 1 animals-12-00498-f001:**
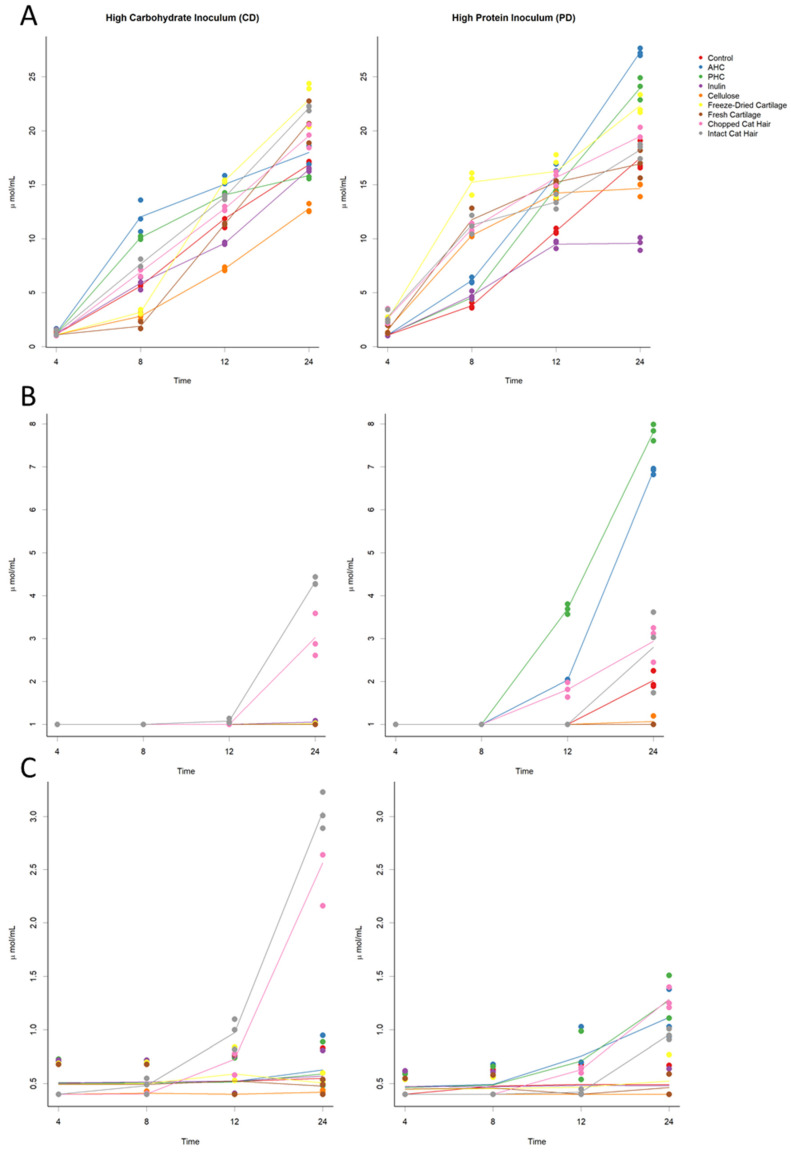
(**A**–**C**). Scatter plots with mean lines of (**A**) acetate (**B**) butyrate and (**C**) propionate concentrations. Time (hours) is along the *x*-axis and organic acid concentration along the *y*-axis (µmol/mL). Each graph depicts both high protein faecal inoculum (PD) and high carbohydrate faecal inoculum (CD) and the changes that occurred over 24 h of fermentation. Each point (coloured circle) represents an individual replicate, and each line shows the mean for each substrate. Control samples are denoted by a red line, AHC (ANZCO hydrolysed collagen) by a blue line, PHC (Peptan hydrolysed collagen) by a green line, cellulose by an orange dashed line, inulin by a purple line, freeze-dried cartilage by a yellow line, fresh cartilage by a brown line, chopped cat hair by a pink line, and intact cat hair by a grey line.

**Figure 2 animals-12-00498-f002:**
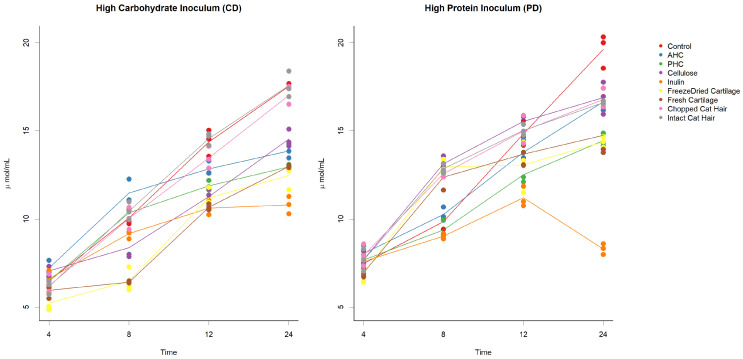
Scatter plots with mean lines of ammonia concentrations. Time (hours) is along the *x*-axis and organic acid concentration along the *y*-axis (mM). Each graph depicts both high protein faecal inoculum (PD) and high carbohydrate faecal inoculum (CD) and the changes that occurred over 24 h of fermentation. Each point (coloured circle) represents an individual replicate, and each line shows the mean for each substrate. Control samples are denoted by a red line, AHC (ANZCO hydrolysed collagen) by a blue line, PHC (Peptan hydrolysed collagen) by a green line, cellulose by an orange dashed line, inulin by a purple line, freeze-dried cartilage by a yellow line, fresh cartilage by a brown line, chopped cat hair by a pink line, and intact cat hair by a grey line.

**Figure 3 animals-12-00498-f003:**
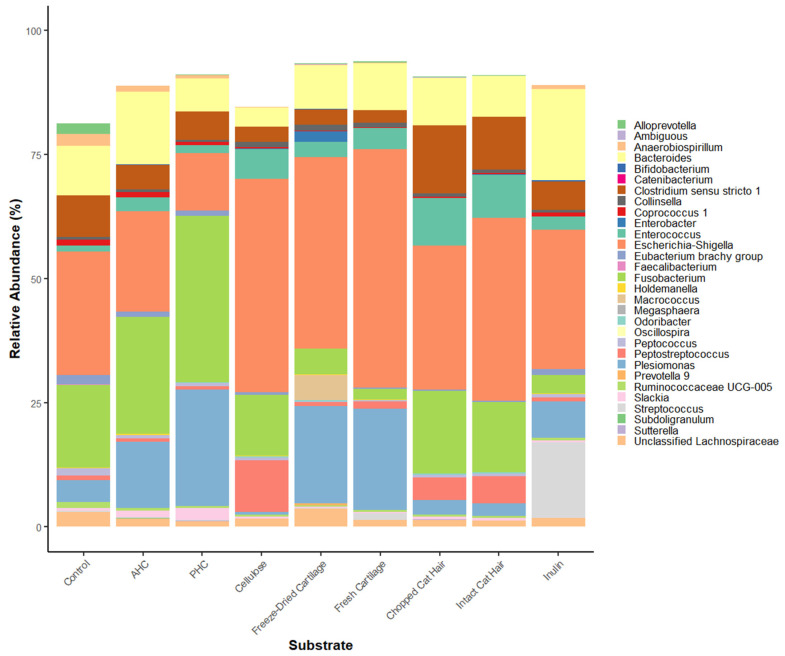
Barplot of the relative abundance of the top 30 bacterial genera present at 24 h in the in vitro fermentation system of the high protein faecal inoculum (PD) according to substrate. Each colour in the bar plot represents a bacterial genus, with the size of the bar denoting their relative abundance.

**Figure 4 animals-12-00498-f004:**
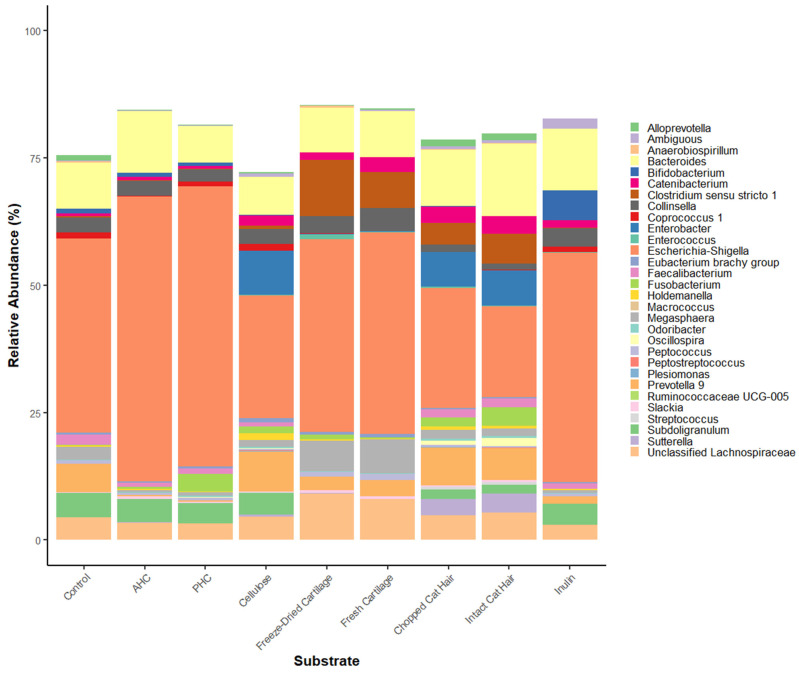
Barplot of the relative abundance of the top 30 bacterial genera present at 24 h in the in vitro fermentation system of the high carbohydrate faecal inoculum (CD) according to substrate. Each colour in the bar plot represents a bacterial genus, with the size of the bar denoting their relative abundance.

**Table 1 animals-12-00498-t001:** Nutrient composition of substrates used for in vitro digestion and fermentation.

	Substrates						
Component	AHC	PHC	Freeze-Dried Cartilage	Fresh Cartilage	Cat Hair	Inulin	Cellulose
Gross energy (kJ/g)	20.58	21.33	22.61	18.31	23.01	16.58	16.65
Ash (% DM)	6.11	0.98	5.08	9.15	1.08	0.1	0.11
Crude Protein (% DM)	89.70 ^¶^	97.48 ^¶^	75.21	69.92	82.20 ^¶^	0.21	0.21
Crude Fat (% DM)	0.54	0.44	13.28	1.69	5.59	2.52	0.11
Crude Fibre (% DM)	0.11	0.11	1.14	0.34	0.75	0.1	65.65
NFE ^1^ (% DM)	3.55	0.98	5.29	18.90	10.38	97.06	33.93
Total Dietary Fibre (% DM)	ND	0.11	12.34	22.03	3.23	†	100
Insoluble Dietary Fibre (% DM)	ND	0.11	5.08	2.37	3.23	†	100
Soluble Dietary Fibre (% DM)	0.86	ND	7.26	19.66	ND	†	ND

^1^ NFE—Nitrogen Free Extract, calculated by difference (100 − crude protein + crude fat + crude fibre + ash). ND—Not detected. † Not assayed. ^¶^ Substrates were converted from nitrogen to protein using the conversion factor 5.55.

**Table 2 animals-12-00498-t002:** Predicted means and associated standard error of the mean of organic acids and ammonia concentrations following in vitro fermentation of each substrate at the 24-h time point in the PD (high protein) faecal inoculum. Letter-based representation of pairwise comparisons is presented. Different letters indicate differences at significance level 0.05, adjusted using the Tukey method. AHC; ANZCO hydrolysed collagen. PHC; Peptan hydrolysed collagen.

	**Acetate (µmol/mL)**			**Butyrate (µmol/mL)**			**Propionate (µmol/mL)**			**Total SCFA (µmol/mL)**		
Substrate	Mean	SEM	Comparison	Mean	SEM	Comparison	Mean	SEM	Comparison	Mean	SEM	Comparison
Control	17.47	0.812	C	2.02	0.114	BC	0.49	0.090	C	30.82	0.362	CD
AHC	27.28	0.196	A	6.90	0.043	A	1.12	0.134	A	45.23	1.223	A
PHC	23.96	0.593	B	7.81	0.111	A	1.28	0.120	A	39.15	0.308	B
Cellulose	14.64	0.372	D	ND	ND	C	ND	ND	C	26.43	2.302	D
Freeze-dried Cartilage	22.33	0.513	B	ND	ND	C	0.52	0.123	BC	38.14	1.288	B
Fresh Cartilage	16.94	0.743	CD	ND	ND	C	0.46	0.063	C	31.77	0.638	CD
Chopped cat hair	19.48	0.469	C	2.94	0.248	B	1.29	0.058	A	33.46	0.980	BC
Intact cat hair	18.22	0.413	C	2.80	0.055	B	0.96	0.029	AB	31.53	1.326	CD
Inulin	9.56	0.343	E	ND	ND	C	0.48	0.080	C	20.09	0.664	E
	**Formate (µmol/mL)**			**Lactate (µmol/mL)**			**Succinate (µmol/mL)**			**Ammonia (µmol/mL)**		
Substrate	Mean	SEM	Comparison	Mean	SEM	Comparison	Mean	SEM	Comparison	Mean	SEM	Comparison
Control	10.71	0.681	AB	2.66	1.260	B	0.673	0.037	C	19.63	0.544	A
AHC	9.81	1.026	ABC	2.04	0.905	B	1.523	0.050	B	16.65	0.392	B
PHC	5.99	0.783	C	1.09	0.430	B	2.310	0.026	A	14.45	0.219	CD
Cellulose	10.22	1.895	ABC	2.15	0.162	B	0.533	0.123	C	16.89	0.525	B
Freeze-dried Cartilage	14.18	0.902	A	3.12	1.412	B	1.493	0.103	B	14.37	0.174	D
Fresh Cartilage	13.27	0.644	AB	2.07	0.886	B	1.300	0.072	B	14.75	0.892	BCD
Chopped cat hair	9.66	0.557	ABC	3.11	0.090	B	1.433	0.126	B	16.78	0.329	B
Intact cat hair	9.45	0.507	BC	1.71	0.448	B	1.180	0.120	B	16.60	0.049	BC
Inulin	8.94	0.762	BC	75.79	8.011	A	1.167	0.044	B	8.30	0.178	E

**Table 3 animals-12-00498-t003:** Predicted means and associated standard error of the mean of organic acids and ammonia concentrations following in vitro fermentation of each substrate at the 24-h time point in the CD (high carbohydrate) faecal inoculum. Letter-based representation of pairwise comparisons is presented. Different letters indicate differences at significance level 0.05, adjusted using the Tukey method. AHC; ANZCO hydrolysed collagen. PHC; Peptan hydrolysed collagen.

	**Acetate (µmol/mL)**			**Butyrate (µmol/mL)**			**Propionate (µmol/mL)**			**Total SCFA (µmol/mL)**		
Substrate	Mean	SEM	Comparison	Mean	SEM	Comparison	Mean	SEM	Comparison	Mean	SEM	Comparison
Control	16.867	0.218276	DE	ND	ND	C	0.55	0.140	B	30.46	1.243959	BCD
AHC	17.99	0.555908	CDE	ND	ND	C	0.63	0.166	B	32.69	2.010075	ABC
PHC	15.85	0.20664	EF	ND	ND	C	0.59	0.152	B	28.1233	0.902244	CD
Cellulose	12.783	0.239188	F	ND	ND	C	0.42	0.012	B	24.24	0.574717	D
Freeze-dried Cartilage	22.873	1.264283	A	ND	ND	C	0.51	0.058	B	37.5167	1.573503	A
Fresh Cartilage	20.767	1.120897	ABC	ND	ND	C	0.48	0.041	B	36.7867	2.106255	AB
Chopped cat hair	19.54	0.62426	BCD	3.027	0.292252	B	2.56	0.214	A	30.4733	1.140663	BCD
Intact cat hair	22.127	0.139084	AB	4.33	0.555108	A	3.04	0.100	A	34.5367	0.398427	ABC
Inulin	16.417	0.096148	DE	1.053	0.027285	C	0.57	0.123	B	31.8	0.831164	ABC
	**Formate (µmol/mL)**			**Lactate (µmol/mL)**			**Succinate (µmol/mL)**			**Ammonia (mM)**		
	Mean	SEM	Comparison	Mean	SEM	Comparison	Mean	SEM	Comparison	Mean	SEM	Comparison
Control	11.933	1.281748	AB	0.783	0.269093	C	0.517	0.048	D	17.533	0.0805	A
AHC	12.963	1.413416	AB	0.563	0.165965	C	0.997	0.033	C	13.859	0.2324	BC
PHC	10.573	1.068431	AB	0.743	0.279543	C	1.320	0.032	C	12.958	0.0869	C
Cellulose	9.937	0.346041	B	1.04	0.280951	C	0.59	0.038	D	14.533	0.2932	B
Freeze-dried Cartilage	12.977	0.302067	AB	5.487	0.086859	B	2.2	0.040	A	12.463	0.41347	C
Fresh Cartilage	14.393	1.083349	A	5.183	1.036345	B	2.287	0.168	A	12.983	0.03876	C
Chopped cat hair	5.243	0.188532	C	2.373	0.315454	C	0.37	0.070	D	17.02	0.4105	A
Intact cat hair	4.937	0.127061	C	2.167	0.384765	C	ND	ND	D	17.578	0.43091	A
Inulin	13.65	0.700595	AB	22.593	0.929737	A	1.717	0.046	B	10.799	0.2829	D

## Data Availability

The datasets generated for this study can be found in NCBI BioProject ID PRJNA767450.

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
