# Peer review of "In Vitro Assessment of Hydrolysed Collagen Fermentation Using Domestic Cat (Felis catus) Faecal Inocula"

_animals, 2022, doi:10.3390/ani12040498_

Round 1

Reviewer 1 Report

Please see attached list of minor recommendations

Author Response

Dear reviewer,

Thank you for your grammatical edits, they have all been incorporated into the manuscript.

Kind Regards,

Christina 

Reviewer 2 Report

The manuscript provided an overview of the effects of the fecal microbiome on collagen fermentation in vitro in cats. This manuscript expanded our knowledge on the capabilities of cat microbiome on breaking down animal-derived fermentable substrates such as collagen which will help in improving diet formulation for domestic cats. The manuscript is well-written. However, I would suggest addressing the comments below before publishing

L97: Add more details about the cat cohort including age, sex, breed … ext

For microbiome analysis: I would suggest adding the following analyses: alpha and beta diversity, predicted metabolic function for the microbiota, clarify which genus were altered in response to each substrate provided. These changes should be tied with the chemical results.

Author Response

Dear reviewer,

Thankyou for your edits, they have been incorporated into the manuscript. Please see attached our point by point response.

Kind regards,

Christina 

Reviewer 3 Report

Dear authors. This is to my opinion a good paper and I have just a few comments you may consider for revision.

Abstract: please add some statistical measures (p-values) to show significance of your main results.

L44 - what is the beneficial level of butyrate? can you give a range or thresholds here?

L60 - SCFA, introduce it when first mentioned in L56.

L86-89 and thereafter - I suggest to use the term microbes instead of bacteria, because not only bacteria are involved in the fermentation processes.

L108-110 - in which way could have the treatment of cartilage affected fermentation due to some pre-decomposition?

L134-136 - which influence may have boiling?

Sec. 2.3. and 2.4. - how many runs did you perform?

Check headline L191.

L224 - "reads" is missing.

Figs 3 and 4 - please show complete y-axis. Are these mean abundances? Please provide information on standard deviation.

L326/327 - please define what is beneficial - the higher the better or is there an optimal range or thresholds?

L379-382 - please explain the sentence - it is not clear what you mean.

L383-385 - yes . . . and how often did you repeat a run/the experiment to obtain real biological replicates?

L386 - use number for the ref.

L388 - pooling of feces should be okay.

Why you did not measure gas production? It is not mandatory but gives information on intensity and duration of microbial activity.

Author Response

(The authors gave the same response as above.)
